# Mitochondria-Targeted Antioxidant SkQ1 Prevents the Development of Experimental Colitis in Mice and Impairment of the Barrier Function of the Intestinal Epithelium

**DOI:** 10.3390/cells11213441

**Published:** 2022-10-31

**Authors:** Artem V. Fedorov, Maria A. Chelombitko, Daniil A. Chernyavskij, Ivan I. Galkin, Olga Yu. Pletjushkina, Tamara V. Vasilieva, Roman A. Zinovkin, Boris V. Chernyak

**Affiliations:** 1Department of Cell Biology and Histology, Biology Faculty, Lomonosov Moscow State University, 119991 Moscow, Russia; 2Belozersky Institute of Physico-Chemical Biology, Lomonosov Moscow State University, 119991 Moscow, Russia; 3Russian Clinical Research Center for Gerontology of the Ministry of Healthcare of the Russian Federation, Pirogov Russian National Research Medical University, 129226 Moscow, Russia; 4HSE University, 101000 Moscow, Russia

**Keywords:** dextran sulfate sodium, SkQ1, Caco-2, intestinal epithelium, ulcerative colitis, reactive oxygen species

## Abstract

Mitochondria-targeted antioxidants have become promising candidates for the therapy of various pathologies. The mitochondria-targeted antioxidant SkQ1, which is a derivative of plastoquinone, has been successfully used in preclinical studies for the treatment of cardiovascular and renal diseases, and has demonstrated anti-inflammatory activity in a number of inflammatory disease models. The present work aimed to investigate the therapeutic potential of SkQ1 and C_12_TPP, the analog of SkQ1 lacking the antioxidant quinone moiety, in the prevention of sodium dextran sulfate (DSS) experimental colitis and impairment of the barrier function of the intestinal epithelium in mice. DSS-treated animals exhibited weight loss, bloody stool, dysfunction of the intestinal epithelium barrier (which was observed using FITC-dextran permeability), reduced colon length, and histopathological changes in the colon mucosa. SkQ1 prevented the development of clinical and histological changes in DSS-treated mice. SkQ1 also reduced mRNA expression of pro-inflammatory molecules TNF, IL-6, IL-1β, and ICAM-1 in the proximal colon compared with DSS-treated animals. SkQ1 prevented DSS-induced tight junction disassembly in Caco-2 cells. Pretreatment of mice by C_12_TPP did not protect against DSS-induced colitis. Furthermore, C_12_TPP did not prevent DSS-induced tight junction disassembly in Caco-2 cells. Our results suggest that SkQ1 may be a promising therapeutic agent for the treatment of inflammatory bowel diseases, in particular ulcerative colitis.

## 1. Introduction

Ulcerative colitis (UC) is a disease of unknown etiology characterized by inflammation of the mucosa and submucosa of the colon and rectum, leading to the development of ulcers [1]. UC is considered a global health problem with an accelerating incidence [2]. The impaired barrier function of the intestinal epithelium and dysregulated immune response in the intestinal mucosa play a key role in the pathogenesis of UC [3]. The integrity of the intestinal barrier depends on antibacterial peptides, immunoglobulin A, mucous layer, and tight junctions (TJs) between intestinal epithelial cells (IEC). TJs are mainly composed of the transmembrane proteins occludin and claudins and the peripheral protein Zonula Ocludence-1 (ZO-1). The disruption of TJs results in the increase of paracellular permeability and penetration of bacteria through the barrier, leading to inflammation [4,5]. Dysfunction of the intestinal epithelium presumably precedes inflammation in the pathogenesis of inflammatory bowel disease (IBD) and colitis. The only anti-inflammatory therapies available include inhibitors of immune cell activation, monoclonal antibodies against inflammatory cytokines, and interferons α/β, which stimulate the production of anti-inflammatory cytokines [3].

Increased oxidative stress has long been recognized as an important pathogenic factor in IBD and colitis, which can impair the intestinal barrier [6]. Signs of oxidative stress are detected both in various experimental models and in patients with IBD. Activated immune cells such as neutrophils produce a predominant amount of reactive oxygen species (ROS) during intestinal inflammation. However, at the initial stages of the disease, IEC-generated ROS may play a key role. Thus, patients with Crohn’s disease and colitis have higher levels of cytoskeletal protein carbonylation and NO production in their intestinal mucosa [7].

In a mouse model of UC induced by sodium dextran sulfate (DSS), oxidative stress was detected using an in vivo electron spin resonance (ESR), or a nitroxyl probe method [8]. It has been shown that intracellular ROS are produced in the initial stage of colitis, and then, in the advanced stage of colitis, ROS are produced predominantly extracellularly. Later, using Overhauser-enhanced magnetic resonance imaging, the same researchers demonstrated ROS production in the epithelium of the distal and proximal colon in the initial stages of DSS-induced colitis before the development of inflammatory changes in the colon mucosa [9]. Ex vivo analysis of colon tissue from DSS-treated mice revealed elevated levels of malondialdehyde and 3-nitrotyrosine, reliable markers of oxidative stress [10].

Mitochondria in the intestine play important roles in immune cell activation, in maintaining barrier integrity, as well as in stemness and differentiation of stem cells [11]. Dysfunction of mitochondria in IEC contributes to the pathogenesis of IBD, including Crohn’s disease and UC. Detailed analysis of mucosal transcriptomes across very large cohorts of pediatric and adult UC patients revealed a significant reduction of mitochondrial metabolic-associated gene expression [12]. Importantly, expression of all 13 genes encoded in the mitochondrial genome was markedly suppressed in UC. In addition, the expression of PGC-1α, the master regulator of mitochondrial biogenesis, was strongly reduced across UC patients. The paralleled analysis of the mitochondrial functions in colonic from UC patients also revealed a significant decline. Mitochondrial unfolded protein response, a universal cell response to severe mitochondrial dysfunction, has been observed in IBD patients’ IEC and UC mouse models [11].

Mitochondrial ROS (mtROS) overproduction is one of the most important consequences of mitochondrial dysfunction. MtROS play a key role in the induction of apoptosis of IEC, leading to the barrier disruption in UC [13]. MtROS are an important component in signaling leading to the activation of nuclear factor-κB (NF-κB) transcription factor [14,15]. NF-κB activation in macrophages and IEC from biopsy samples from patients with IBD, as well as in mouse models of colitis, correlated with inflammation severity [16]. Activation of NF-κB is associated with the production of the pro-inflammatory cytokines in IEC and immune cells, as well as with the disassembly of tight junctions in the intestinal epithelium [17].

Mitochondria-targeted antioxidants (MTA) become a very important tool for studies of mtROS-dependent signaling and are promising candidates for the therapy of various pathologies [18,19]. At the same time, the results of MTA application in models of IBD appear quite controversial (see Section 4. Discussion). Thus, the mitochondria-targeted derivative of ubiquinone (MitoQ) was found protective in the DSS-induced mouse model of UC in the early study [10], but later the opposite effect of MitoQ was reported in the same model [20]. Despite this, the clinical trial of oral MitoQ in moderate ulcerative colitis began in 2021 [21].

The mitochondria-targeted derivative of plastoquinone, SkQ1 (10-(6′-plastoquinonyl) decyltriphenylphosphonium bromide) was shown to be an effective antioxidant in vitro [22] and in vivo [23,24]. It was successfully used in preclinical studies for the treatment of cardiovascular and renal diseases [25,26], and also demonstrated anti-inflammatory activity in acute bacterial infection [27] and the systemic inflammatory response syndrome (SIRS) model [15]. SkQ1 efficiently stimulates compromised dermal wound healing in old and diabetic mice [24,28]. In human neutrophils, SkQ1 prevents oxidative burst, degranulation, and neutrophil extracellular trap (NET) formation caused by various stimuli [29,30]. In endothelial cell culture, SkQ1 has been shown to prevent the disassembly of intercellular contacts [14] and apoptosis [31] induced by TNF. It was shown that SkQ1 prevented TNF-induced activation of NF-κB and expression of adhesion molecules necessary for the adhesion of neutrophil to endothelium during inflammation [14,15].

In the present work, we have shown that SkQ1 at very low doses prevents the development of clinical and histological changes, impairment of the intestinal epithelial barrier, and production of inflammatory cytokines in the DSS-induced mouse model of colitis. In the Caco-2 cell model of the intestinal epithelium, SkQ1 prevented DSS-induced tight junction disassembly.

## 2. Materials and Methods

### 2.1. Mice

Male C57Bl/6 (12 to 15 weeks of age) mice were purchased from the Stolbovaya Research Center for Biomedical Technologies of the Federal Medical and Biological Agency of Russia. Four to five animals were housed per cage. Mice had free access to drinking water and briquetted feed (“Laboratorkorm”, Russia). The experimental protocol was approved by the institutional animal ethics committee (protocol No.1, 6 February 2020).

### 2.2. Induction of Colitis

Mice were divided into four groups: the control group (n = 5), DSS-induced colitis group (n = 9), DSS-induced colitis + C_12_TPP (decyltriphenylphosphonium bromide) group (n = 8), and DSS-induced colitis + SkQ1 (10-(6′-plastoquinonyl) decyltriphenylphosphonium bromide) group (n = 8). Mice received 250 nmol/kg of body weight per day of SkQ1 or C12TPP (synthesized at the Institute of Mitoengineering, Lomonosov Moscow State University) with drinking water for 18 days before induction of colitis and peroral using an automatic pipette during induction of colitis (Appendix A).

For 6 days, mice were given 1% DSS (Dextran sulfate sodium salt, Mr ~ 40,000, Sigma-Aldrich, St. Louis, MO, USA) in their drinking water to induce colitis. Control mice were given tap water. At day 8, all animals received 440 mg/kg of body weight of FITC-dextran (4 kDa, Sigma-Aldrich, St. Louis, MO, USA) and fluorescence was measured in the plasma 4 hours later to evaluate the barrier function of IEC. After isoflurane anesthesia, blood from jugular veins was taken into heparinized tubes. Blood was centrifuged at 2000× *g* for 5 min to collect plasma. The fluorescence of FITC-dextran in plasma was measured on a Fluoroskan Ascent plate fluorometer (Thermo Scientific, Hercules, CA, USA) at a wavelength of 485 nm (extinction wavelength 530 nm).

Clinical scores of colitis were evaluated by the measurement of weight change and colorectal bleeding. Colorectal bleeding was assessed on the following scale: 0—normal stool, 1—soft stools, 2—soft stools with traces of blood, and 3—watery stools with visible rectal bleeding.

The lengths of removed colons were measured in sacrificed mice. The colon was washed and divided into three sections – proximal, medial, and distal. Proximal colon samples were frozen in liquid nitrogen for further mRNA isolation and gene expression analysis. Samples of the distal colon were fixed in 10% neutral buffered formalin for further histological analysis.

### 2.3. Histological Analysis

After fixation, the colon samples were processed for paraffin embedding, sectioned at 5 µm thickness, and stained with hematoxylin and eosin. Histological scoring was performed as previously described [32]. The histological grading of colitis is described in Appendix A. 

### 2.4. Reverse Transcription-PCR

RNA from the proximal colonic section was isolated using a QIAGEN RNeasy Blood and Tissue kit according to the manufacturer’s instructions. The integrity and purity of RNA were verified by agarose gel electrophoresis and spectrophotometry using a NanoDrop ND-1000. RNA was reverse transcribed by SuperScript III reverse transcriptase (Thermo Fisher Scientific, Waltham, MA, USA). Quantitative PCR (qPCR) was carried out with EvaGreen intercalating dye (Syntol, Moscow, Russia) on the CFX96 Touch™ Real-Time PCR Detection System (BioRad, Hercules, CA, USA). PCR conditions included a preheating step at 95 °C for 3 min and 40 cycles at 95 °C for 15 s, 57 °C for 15 s, and 72 °C for 15 s coupled with fluorescence measurement. The amplification specificity was confirmed by melt curves analysis. Each sample was run in triplicate, and a non-template control was added to each run. The choice of reference genes TATA-box-binding protein (Tbp) and eukaryotic translation elongation factor 2 (Eef2) was based on the results of a stability study of the reference genes in a mouse model of DSS-induced colitis [33]. The expression levels of the target genes were determined by the ΔΔCt method with the calculated primer efficiencies and normalized on the geometric mean of selected reference genes. Primer sequences are listed in Appendix A.

### 2.5. Cell Culture

Caco-2 cells (ATCC HTB-37) were cultivated in the DMEM medium containing 10% fetal bovine serum (FBS) at 37 °C and 5% CO_2_. The cells were seeded on glass coverslips in a 6-well plate (100,000 cells/well). After cell adhesion, SkQ1 and C_12_TPP were added to the final concentration of 2 nM. This concentration was found to be the most effective in our previous work [15]. Two days after incubation, the cells were treated with 2% DSS for two days.

### 2.6. Fluorescence Microscopy

Cells were washed with DMEM, fixed with a 2% solution of paraformaldehyde in DMEM for 10 min at 37 °C, and permeabilized with 0.2% Triton X-100 in PBS for 5 min at RT. Cells were stained with anti-ZO-1 polyclonal antibodies (AB2272, Merck Millipore, Billerica, MA, USA) overnight at +4 °C. The cells were stained for 1 h, at RT with secondary cross-adsorbed Goat anti-Rabbit IgG (H+L) antibodies, conjugated with Alexa Fluor 488 (A11034 Thermo Fisher Scientific, Waltham, MA, USA), phalloidin-TRITC (Sigma-Aldrich, St. Louis, MO,), and Hoechst 33342 (Biotium, Fremont, CA, USA). After washing with PBS, the cells were embedded using Aqua-Poly/Mount (Polysciences, Warrington, PA, USA). Microscopy was performed using a fluorescent microscope Axiovert 200M (Zeiss, Oberkochen, Germany) with objective EC Plan-NEOFLUAR 63x/0,75 Ph2 (Zeiss, Oberkochen, Germany). Analysis of morphological data was performed in ImageJ. The percentage of cells with intact tight junctions was estimated. In four separate experiments, 500 to 600 cells were counted. Cells with a continuous line of ZO-1 around their perimeter were identified as cells with intact tight junctions.

### 2.7. Statistical Analysis

ANOVA data analysis with post-hoc Dunnett’s multiple comparison test was performed using the GraphPad Prism 8.0 software (GraphPad Software, San Diego, CA, USA). The DSS-treated group was set as a control group for Dunnett’s post-hoc tests. Data are presented as mean ± SD. *p*-values less than 0.05 (*), 0.01 (**), 0.001 (***), and 0.0001 (****) were considered significant.

## 3. Results

### 3.1. SkQ1 Prevents the Development of Dextran Sulfate Sodium-Induced Colitis in Mice

In mice treated with DSS, clinical signs of colitis such as weight loss and bloody stool were observed. The development of DSS-induced colitis was accompanied by a significant decrease in body weight (Figure 1A,B) and colorectal bleeding (Figure 1C), which were prevented by SkQ1. Treatment of mice by C_12_TPP, the analog of SkQ1 lacking the antioxidant quinone moiety, using the same scheme did not protect against colitis (Figure 1).

An important sign of the development of UC is a decrease in the length of the colon. The severity of UC is determined by the reduction of colon length. In agreement with this, DSS-treated mice had significantly reduced colon length compared with control mice (Figure 1D,E). SkQ1 prevented the development of this trait, while C_12_TPP had no protective effect (Figure 1D,E).

An increase in intestinal epithelium permeability is another critical parameter in UC. A significant increase in mouse intestinal permeability for FITC-dextran was observed in DSS-treated mice compared with control mice (Figure 1F). In mice treated with SkQ1, there was a significant decrease in intestinal permeability compared with a group with colitis, while C_12_TPP had no noticeable effect (Figure 1F).

The results of histological examination of the distal colon in mice treated with DDS confirmed the development of colitis. An almost complete absence of crypts and epithelial lining of the mucosa (mucous membrane) was observed. In the lamina propria of the mucous membrane, instead of crypts, there was granulation tissue with dilated blood vessels and a pronounced inflammatory infiltrate. Edema and inflammatory infiltration were also observed in the submucosal layer (submucosa). In mice treated with SkQ1, the microscopic structure of the intestine did not significantly differ from that of intact control animals, while C_12_TPP had no protective effect (Figure 2).

The development of the inflammatory process in the colon in mice with DSS-induced colitis was confirmed by the results of the analysis of pro-inflammatory cytokine gene expression. Thus, a significant increase in mRNA of TNF, IL-6, IL-1β, and ICAM-1 genes in the proximal colon in mice with DSS-induced colitis compared with control mice (Figure 3). In full agreement with its protective effect, SkQ1 prevented an increase in the expression of all these genes, while C_12_TPP had no effect. No significant difference was found in the expression of IL-1б and IL-18. An increase in heme oxygenase-1 (HO-1) gene expression was observed, and that SkQ1, but not C_12_TPP, prevented this increase (Figure 3). HO-1 expression is a reliable marker of Nrf2 activation, so these data suggest that DSS induces NRf2 activation due to the action of mtROS.

### 3.2. SkQ1 Prevents the DSS-Induced Disassembly of Intercellular Contacts in the Caco-2 Cells

To test the hypothesis that the therapeutic effect of SkQ1 may be explained by its direct effect on the barrier function of the intestinal epithelium, we evaluated the SkQ1 effect on DSS-induced disassembly of intercellular contacts in the Caco-2 intestinal epithelial cells. Disassembly of the contact structure was assessed by the relocation of TJ ZO-1 protein from the contact area to the cytoplasm. Pretreatment of cells with SkQ1 suppressed the release of ZO-1 from the contacts, while C_12_TPP had no effect (Figure 4). In addition, we observed an increase in the number of actin filaments resembling stress fibers in cells treated with DSS. Such fibers are usually formed in response to mechanical and other stresses. SkQ1, but not C_12_TPP, prevented the formation of these fibers (Figure 4).

## 4. Discussion

Oral administration of SkQ1 prevented the development of morphological and clinical signs of UC pathology in DSS-induced mice (Figure 1). SkQ1 also prevented an increase in intestinal epithelial barrier permeability (Figure 1D) and the expression of inflammatory mediators in the proximal colon of DSS-treated mice (Figure 2). Histological changes confirmed the prevention of pathological changes in the colon by SkQ1 (Figure 3). The control compound C_12_TPP did not affect the development of pathologies, thus it can be assumed that the scavenging of mtROS mediates the therapeutic action of SkQ1. Previous studies have shown that SkQ1 is an effective antioxidant in vitro [22] and in vivo [23,24].

DSS-induced colitis is a widely used mouse model of UC. In fact, MitoQ was the first MTA found to be protective in this model [10]. Oral administration of MitoQ for 7 days after UC induction reduced clinical and histological changes in the colon and suppressed activation of pro-inflammatory cytokines IL-1β and IL-18. The production of these cytokines is predominantly dependent on the activation of NLRP3 inflammasome in macrophages, so it was assumed that these cells are the main target of MitoQ [10]. However, the opposite effect of MitoQ on macrophage activation was later reported using the same model [20]. MitoQ has been shown to reduce oxidative stress and NF-κB activation but exacerbated colitis, presumably by inhibiting macrophage polarization to the anti-inflammatory M2 phenotype. In both studies, the integrity of the intestinal epithelial barrier was not analyzed. As shown in Figure 3, SkQ1 inhibited the expression of immature IL-1β, as well as the inflammatory cytokines TNF and IL-6, and adhesion molecule ICAM-1. Since NF-κB controls the expression of all these genes, activation of NF-κB by mtROS is proposed in the DSS-induced UC model. This assumption is in good agreement with our earlier studies, which showed the prevention of TNF-induced NF-κB activation by SkQ1 was shown in the endothelium [14,15].

More recently, two independent studies have shown that the effect of MitoQ in ischemic [34] and lipopolysaccharide-induced [35] mouse models of UC was mediated by activation of nuclear factor erythroid 2-related factor 2 (Nrf2). Nrf2 is a transcription factor up-regulated by oxidative and electrophilic stresses to activate the expression of antioxidant and other protective genes. The protective effect of Nrf2 in IBD is well known [36]. Activation of Nrf2 by MitoQ can be explained by the pro-oxidant activity of this MTA inherent in quinone-based antioxidants [37]. Antioxidant enzymes controlled by Nrf2 can protect mitochondria as well as other cellular structures from cytosolic ROS, so the role of mtROS scavenging by MitoQ in protection against UC remains unclear. As shown in Figure 3, DSS stimulated and SkQ1 prevented the expression of hemoxygenase-1 (HO-1), a strong marker of Nrf2 activation [38]. These data assume that Nrf2 activation in the DSS-induced model of UC is mtROS dependent. Importantly, SkQ1 was used at a significantly lower daily dose (0.25 µmol/kg) in our study than MitoQ was (6–80 µmol /kg) in the cited studies. Furthermore, SkQ1’s pro-oxidant activity is approximately 100 times lower than that of MitoQ [39]. The low pro-oxidant activity of SkQ1 may explain the lack of additional Nrf2 activation in the DSS-induced UC model.

Another MTA, MitoTEMPO, has been shown to reduce intestinal barrier dysfunction as assessed by Escherichia coli transepithelial flux in the DSS-induced mouse model of UC [40]. The possible activation of Nrf2 was not analyzed in this study, so the detailed mechanism of action of MitoTEMPO remains unclear. More recently, MitoTEMPO has been shown to protect against IBD in mice with an inducible deletion of prohibitin 1 in intestinal epithelial cells [41]. Prohibitin 1 is a chaperone for mitochondrial DNA, which expression is downregulated during active human IBD and DSS-induced colitis in mice [42], so its possible role in the pathogenesis of IBD may be mediated by excessive production of mtROS. MitoTEMPO does not rapidly regenerate in mitochondria, so unlike SkQ1 and MitoQ, it cannot act as a “rechargeable” antioxidant. This is one of the reasons why MitoTEMPO is used at relatively high doses in cellular or animal models, and no clinical trials have been reported with it.

As shown in Figure 4, SkQ1 prevents tight junction disassembly in Caco-2 cells, suggesting that mtROS regulates this type of cell junction. We measured the total level of ROS in the cells using DCFH2-DA but found no significant changes after DSS treatment (Appendix A). Moreover, we did not register any changes induced by SkQ1 in either control or DSS-treated cells. These data suggest that the Caco-2 model of DSS treatment may not be completely relevant to the animal colitis model. The latter involves a complex interaction of many cell types and supposedly has prominent ROS induction. Thus, our suggestion is based only on previous studies that clearly demonstrate that SkQ1 effectively removes mtROS in a variety of animal models and cell types [22,23,24]. We hypothesize that the increase in mtROS induced by DSS may be either small or transient, so our methods were unable to detect this response. Loss of ZO-1 from TJ complexes and increased permeability of the intestinal epithelium have been shown to precede the development of significant intestinal inflammation in DSS-induced colitis [43]. Disruption of the TJ complex probably underlies the development of the inflammatory infiltrate observed in colitis. It has been previously shown that oxidative stress induced by moderate hypoxia followed by reoxygenation destroys TJ complexes in the blood-brain barrier [44]. Exogenous superoxide-producing xanthine oxidase [45] as well as acetylsalicylic acid induce ROS-dependent TJ disassembly in Caco-2 cells [46]. In Caco-2 cells treated with inflammatory cytokines, nitrosative stress associated with ROS overproduction can also contribute to TJ disruption [47]. The possible role of mtROS in these models has not been studied. MitoQ and MitoTEMPO have been shown to protect against TJ degradation induced by osmotic stress and DSS in Caco2, but important experiments with control TPP-based compounds were not presented [48].

Disassembly of the TJ complex may be dependent on actin cytoskeleton reorganization [49], particularly the formation of contractile actin stress fibers [50]. On the other hand, activation of actomyosin contractility of stress fibers improves TJ assembly and epithelial barrier function [51,52]. The formation of stress-like fibers was induced by DSS in Caco-2 cells and was prevented by SkQ1 (Figure 4), indicating that mtROS play a key role in the reorganization of the actin cytoskeleton. The possible role of these actin fibers in TJ disassembly and disruption of the integrity of the intestinal epithelial barrier requires further study.

Clinical trials with traditional dietary antioxidants in IBD patients gave controversial results. In one of the randomized controlled trials, it was shown that oral supplementation of the antioxidants resulted in significant clinical improvement in patients with colitis and a decreased requirement for corticosteroids, indicating that oxidative stress may have a causative role in the disease [53]. By contrast, other randomized controlled trials did not reveal an effect of antioxidant vitamins C and E on the disease progression while the indices of oxidative stress significantly decreased [54]. The reasons for this discrepancy remain unknown, but probably reflects the differences in dosage and potency of antioxidants. 

Mitochondria-targeted antioxidants, such as SkQ1, MitoQ, and MitoTEMPO demonstrated very high efficiency in preclinical studies of various inflammatory pathologies [19]. The high efficiency of eye drops containing SkQ1 has been demonstrated not only in various models of inflammatory eye diseases in animals [55], but also in a clinical study of dry eye syndrome [56]. The clinical trial of oral MitoQ in moderate ulcerative colitis was started in 2021 [21]. The findings of this investigation point to SkQ1 as a potentially effective therapeutic agent for the management of inflammatory bowel diseases, particularly ulcerative colitis.

## 5. Conclusions

This work confirms the potential of mitochondria-targeted antioxidants for the treatment of inflammatory diseases, including IBD and colitis. SkQ1 consumption had a profound protective effect on the colon of DSS-treated mice. The underlying mechanism of protective SkQ1 action includes the preservation of cell-to-cell contact integrity in vitro. Our study provides additional proof for the investigation of the mitochondria-targeted antioxidants in future clinical trials. Thus, mitochondrial ROS inhibition is a potential approach for the prevention and treatment of IBD.

## Figures and Tables

**Figure 1 cells-11-03441-f001:**
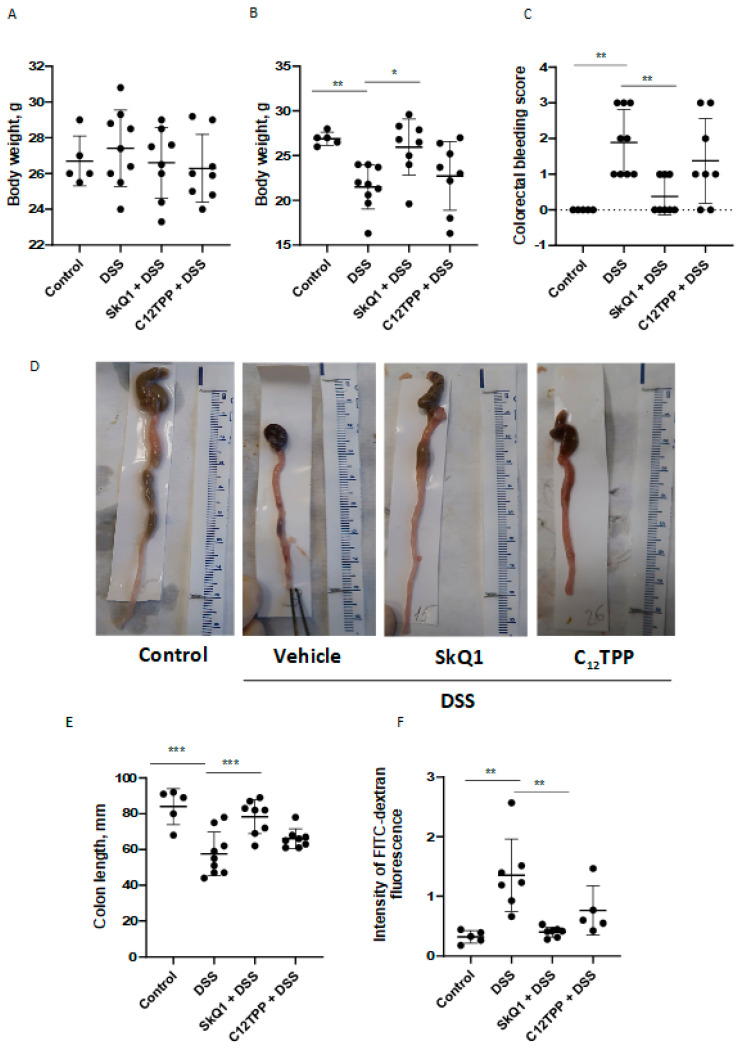
Effect of SkQ1 and C_12_TPP on the development of DSS-induced colitis in mice. (**A**) Body weight of mice on day 18 before colitis induction. (**B**) Body weight of mice on day 24 after colitis development. (**C**) Results of assessment of colorectal bleeding score on day 26. (**D**) Representative images of freshly removed colons. (**E**) The results of measurement of colon length. The units are mm. (**F**) Permeability of the intestinal epithelium, measured by the fluorescence of FITC-dextran in blood plasma. Animal groups (n = 5–9) were designed as follows: “Control”—untreated animals; “DSS”—DSS-treated mice; “SkQ1 + DSS”—SkQ1 treated DSS mice; “C_12_TPP + DSS”—C_12_TPP-treated DSS mice. Results are expressed as mean ± SD (n = 5–9). * *p* < 0.05, ** *p* < 0.01, *** *p* < 0.001 as compared with the DSS-treated mice.

**Figure 2 cells-11-03441-f002:**
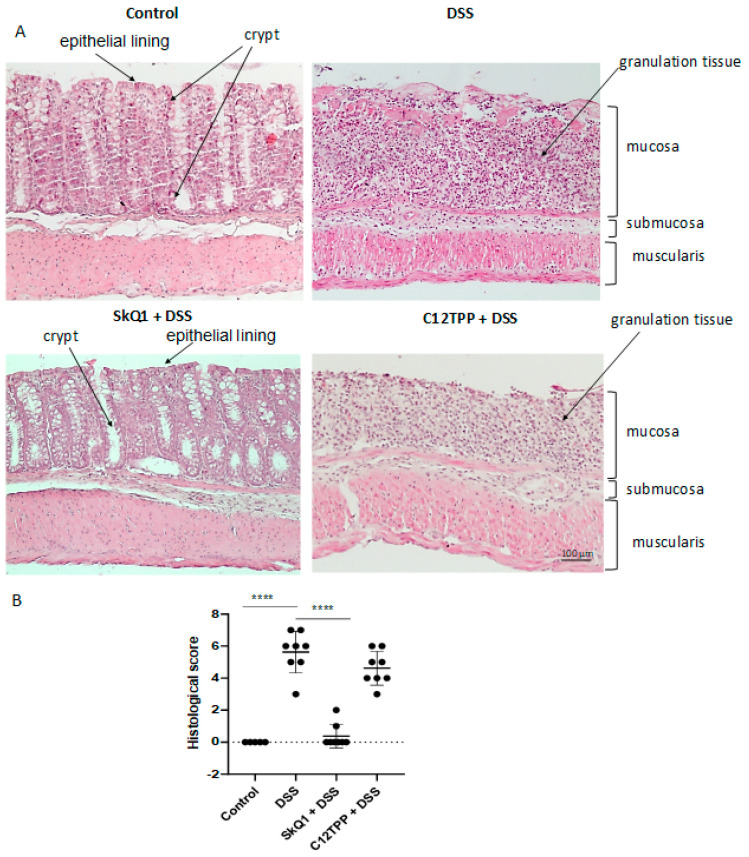
Effect of SkQ1 and C_12_TPP on the microscopic structure of the distal colon in mice with DSS-induced colitis. Animal groups are indicated in Figure 1. (**A**) Representative sections of the distal colon stained with hematoxylin and eosin. (**B**) Results of histological assessment of the extent of damage to the colon. Results are expressed as mean ± SD (n = 5–9). **** *p* < 0.0001 as compared with the DSS-treated mice.

**Figure 3 cells-11-03441-f003:**
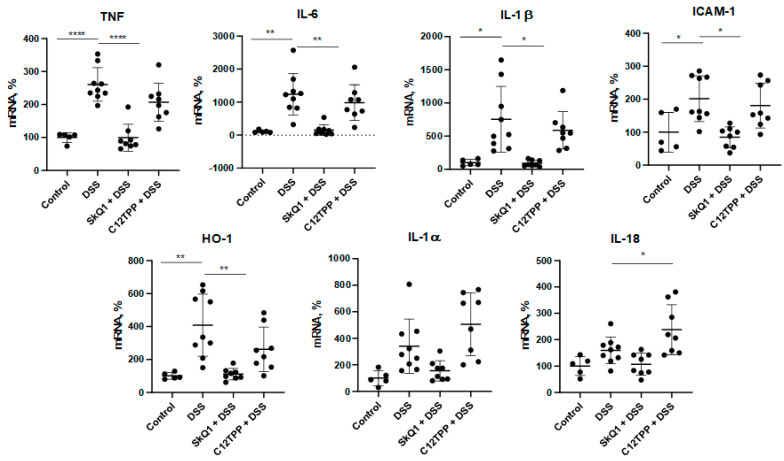
Effect of SkQ1 and C_12_TPP on gene expression of pro-inflammatory molecules in the proximal colon in mice with DSS-induced colitis. Animal groups are indicated in Figure 1. Results are expressed as mean ± SD (n = 5–9). * *p* < 0.05, ** *p* < 0.01, **** *p* < 0.0001 as compared with the DSS-treated mice.

**Figure 4 cells-11-03441-f004:**
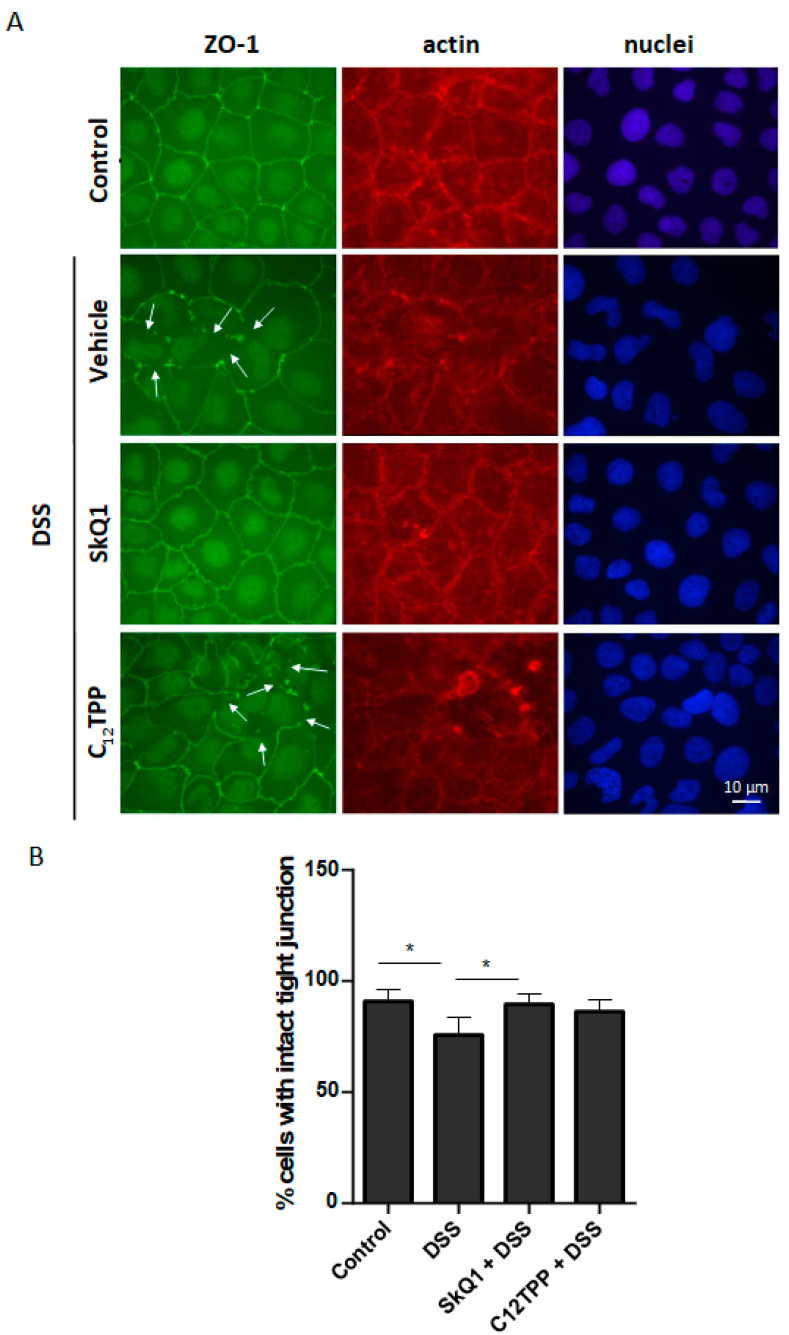
Effect of SkQ1 and C_12_TPP on tight junction disruption in Caco-2 cells after DSS exposure. (**A**) Immunofluorescence of the tight junction (green), actin cytoskeleton (red), and nuclei (blue) in cells. After treatment with 2 nM of SkQ1 or C_12_TPP for 48 h, cells were treated with 2% DSS for 2 days. The cells were fixed with 2% paraformaldehyde and stained with anti-ZO-1 antibody (ZO-1) with phalloidin-TRITC (actin), and Hoechst 33342 (nuclei). (**B**) The percentage of cells with intact tight junctions. The arrows indicate the disassembly of tight junctions (lack of ZO-1 between cells). Between 500 to 600 cells were analyzed in four independent experiments. Results are expressed as mean ± SD. * *p* < 0.05 as compared with the DSS-treated cells.

## Data Availability

Not applicable.

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
