# Peer review of "Mitochondria-Targeted Antioxidant SkQ1 Prevents the Development of Experimental Colitis in Mice and Impairment of the Barrier Function of the Intestinal Epithelium"

_cells, 2022, doi:10.3390/cells11213441_

Round 1

Reviewer 1 Report

The manuscript by Fedorov et al. describes the efficacy of SkQ, a mitochondria-targeted antioxidant, in preventing sodium dextran sulfate-induced colon lesions. The study is well designed, it includes both animal and cellular models. The results are novel and interesting, but lack some technical aspects and details; the study can be potentially accepted after a major revision. 

Major remarks:

  1. Data representation is one of the major remarks for this study. Figure 1A  - matched results (animal before intervention and after intervention) of body weight are preferred for this type of data, please, update the representation for each group. If body weight monitoring was performed daily, its daily progression in each group can be also shown. Please, provide representative figures for the results shown on Figure 1B (representative images), 1C (representative images), 1D (representative flow cytometry graphs (?), see below). Also, it is preferable to show on the graph individual points that correspond to each animal (biological replicate), instead of only bar plots. Please, also provide a scheme of animal studies (experimental design for sodium dextran sulfate-induced colitis in mice).

  2. Some details are missing in the Materials and Methods Section. Please define how assessment of colorectal bleeding was done and how the score is defined. Please, describe how FITC-dextran fluorescence was measured in the serum. Why is it described in the figure 1D legend as "optical density"? Please, provide the details how data presented on Figure 4 (the percentage of cells with intact tight junctions) were analyzed: how cells with intact tight junctions were defined? Was it a manual or semi-automatic screen? Which cells were included or excluded from the analysis? etc. Additionally, in general, please, provide more informative legends for the presented figures and panels. 

  3. The results on Figure 3 can be supported by the analysis of protein expression in addition to mRNA expression.

  4. While the authors aim to analyze the effect of antioxidants on a colitis model and find SkQ as a potentially effective compound, the confirmation of direct SkQ effect in decreasing ROS production in their colitis model (either animal or cellular) is missing. This could be added to the study to confirm the mechanism of SkQ action in the used model and support their conclusions. For example, the authors state that:

"The formation of stress-like fibers was induced by DSS in Caco-2 cells and was prevented by SkQ1 (Fig. 4), indicating that mtROS play a key role in the reorganization of the actin cytoskeleton." mtROS, however, were not analysed.

Minor remarks:

The Introduction section could be shortened to make it more focused and related to the topic of research.

Please, indicate the set level of significance in Statistical analysis section

L. 198 - Figure 1A is not cited in the text

Reviewer 2 Report

Fedorov and colleagues tried to investigate the therapeutic potential of SkQ1 in the prevention of sodium dextran sulfate experimental colitis and impairment of the barrier function of the intestinal epithelium in mice.

- The abstract lacks the methods part. There is no details about the model of induction of ulcerative colitis, drugs used in treatment, pathological investigations or protein and gene expression.

- The introduction is too long and contain many details.

- In the methods part, did you treat mice by adding the two investigated materials to food or drinking water or you used oral gavage?

- Try to organize your results under subtitles to be easily followed.

- In figure 2, use arrows to indicate the pathological changes in micro-images.

- In the results and discission, you had many parts about Nrf2 expression. However, none of your figures showed results for Nrf2.

- The supplementary table 1, had the primer sequence for TBP. However, none of your results or discussion showed this gene.

- In the conclusion part, add the clinical significance of your discoveries.

Reviewer 3 Report

To the authors:

1.     Although the use of SkQ1 clearly demonstrated anti-inflammatory, improved barrier integrity, and reduce disease activities with the four groups (control, DSS alone, SkQ1 + DSS, and C12TPP + DSS), it would be more insightful to add C12TPP alone and SkQ1alone treatment. These would serve as additional control groups.

2.     In the in vivo studies, only male mice were used. Including female mice would make these studies more inclusive in regard to the potential therapeutic effect of SkQ1.

3.     Caco-2 cells are known to differentiate to absorptive type of cell, and their use in this study is appropriate. However, replicating the barrier FITC-dextran study on another cell line, such as HT29-MTX (differentiate to secretory cells) would make the study stronger.

4.     Representative images showing changes in colon length would be useful for Figure 1c.

5.     Including arrows to show infiltration of immune cells into the colonic mucosa would be helpful.

6.     Histological assessment was shown from the distal colons in Figure 2. However, the mRNA expression of pro-inflammatory cytokines (Figure 3) was performed from proximal colon. Is there any particular reason for this inconsistency? Otherwise, include data for Figures 2 and 3 from both regions of the colon.

7.     Since expression of NF-kB is implicated in the induction of pro-inflammatory cytokines reported in Figure 3, it would be informative to include mRNA expression levels of NF-kB.

8.     Arrows indicating the relocation of TJ ZO-1 protein would be helpful to visualize in Figure 4. In addition, including mRNA expression of tight junction proteins, such as occludin and claudin would confirm barrier integrity.

9.     Sentence in line 82 needs reorganization.

10. Does the sentence in line 184 need to read as…..-4oC?

11. Sentence in line 198 should be (Figure 1A and B).

12. Sentences in lines 235, 237, and 252: DSS not DDS.

13. The mtROS scavenging effect of SkQ1 was reported in previous studies and could be sited to support the statement in the discussion section (line 267)

14. It was stated in the discussion section (lines 305-306) that the function of Prohibitin 1 is unknown. However, that is not true. It is well known that Prohibitin 1 is important for cell cycle, apoptosis, and transcription. In the intestinal cells, Prohibitin 1 is essential for chaperon function and to stabilize mitochondrial DNA (https://gut.bmj.com/content/69/11/1928).

Round 2

Reviewer 1 Report

The authors revised the manuscript according to reviewers' remarks, which significantly improved the data representation and overall manuscript quality. I have some minor remarks that should be corrected before publication.

Minor remarks:

The authors measured the level of ROS in their cellular model using DCFH2-DA and mtROS level using MitoSOX but found no significant changes, I suggest to provide these data anyway (in the main text or supplementary). These data might indicate that the cellular model of DSS treatment is not completely relevant to the animal colitis model, where they supposedly have ROS induction, which should be clearly stated as study limitation.

Figure 3 is out of page borders.

L162 -  "()-treated group" - a typo?

Figure 1C - it is unclear to which comparison the second ** refers to (no between-group line provided)

Figure 1F - please, explain why the data for some animals (biological replicates) are missing

Figure 2B - please, use the same graph type as in Figure 1 to show individual values for each animal (biological replicate)

Figure 3 "IL-18"- the upper whisker for DSS group is not visible

L. 409-411 " Thus our suggestion is based only on previous studies that clearly demonstrate that SkQ1 effectively removes mtROS in a variety of animal models and cell types." - please, provide the references for these previous studies.

Paragraph - 2.6. Statistical analysis - Please, indicate which group was set as a control group for Dunnett's post-hoc tests.

Reviewer 2 Report

No more comments 

Author Response

We are grateful to Referee for the appreciation of our work.

Reviewer 3 Report

Dear Authors,

Thank you for making some of the recommended changes and also provided explanations for my comments.

Author Response

We would like to thank you again for your valuable comments and appreciation of our work..